# LILRB4 Checkpoint for Immunotherapy: Structure, Mechanism and Disease Targets

**DOI:** 10.3390/biom14020187

**Published:** 2024-02-04

**Authors:** Zhiqing Xiang, Xiangli Yin, Leiyan Wei, Manqing Peng, Quan Zhu, Xiaofang Lu, Junshuang Guo, Jing Zhang, Xin Li, Yizhou Zou

**Affiliations:** Department of Immunology, Xiangya School of Medicine, Central South University, Changsha 410078, China; 216511061@csu.edu.cn (Z.X.); yinxiangli@csu.edu.cn (X.Y.); 216511057@csu.edu.cn (L.W.); 226511061@csu.edu.cn (M.P.); zhuquan_csu@csu.edu.cn (Q.Z.); xiaofanglu@csu.edu.cn (X.L.); 226502018@csu.edu.cn (J.G.); 226501024@csu.edu.cn (J.Z.); 226501023@edu.cn (X.L.)

**Keywords:** LILRB4, checkpoint, immune tolerance, tolerogenic cell

## Abstract

LILRB4, a myeloid inhibitory receptor belonging to the family of leukocyte immunoglobulin-like receptors (LILRs/LIRs), plays a pivotal role in the regulation of immune tolerance. LILRB4 primarily mediates suppressive immune responses by transmitting inhibitory signals through immunoreceptor tyrosine-based inhibitory motifs (ITIMs). This immune checkpoint molecule has gained considerable attention due to its potent regulatory functions. Its ability to induce effector T cell dysfunction and promote T suppressor cell differentiation has been demonstrated, indicating the therapeutic potential of LILRB4 for modulating excessive immune responses, particularly in autoimmune diseases or the induction of transplant tolerance. Additionally, through intervening with LILRB4 molecules, immune system responsiveness can be adjusted, representing significant value in areas such as cancer treatment. Thus, LILRB4 has emerged as a key player in addressing autoimmune diseases, transplant tolerance induction, and other medical issues. In this review, we provide a comprehensive overview of LILRB4, encompassing its structure, expression, and ligand molecules as well as its role as a tolerance receptor. By exploring the involvement of LILRB4 in various diseases, its significance in disease progression is emphasized. Furthermore, we propose that the manipulation of LILRB4 represents a promising immunotherapeutic strategy and highlight its potential in disease prevention, treatment and diagnosis.

## 1. Introduction

Immune tolerance is a state of specific nonresponse exhibited by immunologically active cells when exposed to antigenic substances. In diseases such as autoimmune diseases and transplantation immune tolerance, enhanced immune tolerance is often required to control the disease process [1]. In the past, immunosuppressive drugs have been used to create an immunosuppressive microenvironment in the body, but their long-term use can lead to a significant increase in the risk of infections and even malignant tumors in patients [2]. Therefore, there is an urgent need to find a therapeutic approach that specifically induces immune tolerance, targeting only the pathogenic immune response without compromising the protective host immune response [3]. One of the most discussed ways to specifically induce immune tolerance is through the modulation of immune checkpoint proteins. Immune checkpoint proteins act as regulatory switches for the immune system and can help maintain immune homeostasis; thus, immune responses can be selectively suppressed via rational manipulation of immune checkpoints to induce immune system tolerance to specific antigens [4].

Immune checkpoint receptors are a class of immunosuppressive protein receptors expressed on immune cells that modulate the extent of the immune response [5]. It has been found that activation of immune checkpoint proteins strengthens the body’s immune tolerance response, resulting in exertion of a powerful immunosuppressive effect, thereby effectively suppressing disease progression. Since the activation of most immune checkpoints is initiated by ligand–receptor interactions, they are easily blocked by antibodies or modulated by recombinant forms of ligands or receptors [6]. Activating cellular immunity by blocking immune checkpoints can have a therapeutic anti-tumor effect, and this therapeutic modality has been shown to produce long-lasting clinical responses. On the contrary, activation of immune checkpoint proteins usually causes suppression of the bodily immune response and induces the onset of immune tolerance, which can play a key role in treating autoimmune disorders or inducing transplantation immune tolerance, among other disease processes [7,8,9,10]. Therefore, immune checkpoint proteins are expected to play a unique role in disease treatment as part of new immunotherapeutic strategies.

Leukocyte immunoglobulin-like receptor B (LILRB) and immune checkpoint proteins such as cytotoxic T lymphocyte-associated protein 4 (CTLA4) and programmed cell death protein 1 (PD-1) share the same inhibitory ITIM motif and are novel immune checkpoint proteins [11,12,13]. The LILRB family is a family of transmembrane glycoproteins with immunoglobulin (Ig-like) structural domains in the extracellular region and a tyrosine-based immune receptor inhibitory motif ITIM in the intracellular region, which exerts immunosuppressive effects by inhibiting downstream signaling pathways via the enzymatic phosphatase action of protein tyrosine phosphatase1/2 (SHP1/2) proteins [14,15,16]. In this family, LILRB4 is well known for its role in promoting the migration and invasion of leukemia cells and inhibiting the proliferation of T cells [17]. Indeed, under physiological conditions, LILRB4 is expressed on a wide range of immune cells and thus in the immune system, where it also plays an important role. In addition to tumors, differential expression of LILRB4 is present in a variety of immune system diseases, such as Kawasaki disease (KD) and systemic lupus erythematosus (SLE) [18,19,20]. The important role of LILRB4 in the immune system and its differential expression in a variety of diseases makes it a potential therapeutic target for a wide range of diseases.

## 2. Structure and Expression of LILRB4 and Its Ligand Molecules

The leukocyte immunoglobulin-like receptor (LILR) family comprises 11 immunomodulatory receptors encoded in the leukocyte receptor complex on chromosome 19. These they are key regulators of leukocyte function [21]. Members of the leukocyte immunoglobulin-like receptor family include two classes: activating LILRs (LILRA1-6) and inhibitory LILRs (LILRB1-5) [15,22].

Inhibitory LILRs (LILRBs) are predominantly expressed on leukocytes, and LILRBs can trigger inhibitory signaling through the long cytoplasmic tails of their intracellular immunoreceptor tyrosine inhibitory motifs (ITIM) [23] (Figure 1). These ITIM inhibitory motifs utilize the enzymatic phosphatase action of SHP1/2 proteins to inhibit downstream signaling pathways, such as protein kinase B (AKT) and extracellular signal-regulated kinase (ERK), for further immunosuppressive effects [24,25]. SHP1/SHP2 have also been reported to regulate the activation of the Janus kinase and activator of transcription signaling pathway (JAK/STAT pathway), which plays a role in cell maturation and immune function regulation [26]. Meanwhile, ITIM-dependent SHP1/SHP2 recruitment to LILRB has broad implications for the inhibition of the Syk/Src signaling cascade associated with immune activation [15].

LILRB4 is a member of the LILR immunoglobulin superfamily, which is highly conserved within species. This receptor molecule shares 97% genetic homology between humans and in mice [27]. In humans, the LILRB4 gene is localized to the leukocyte receptor cluster on chromosome 19q13.4 and encodes a leukocyte immunoglobulin-like receptor consisting of two extracellular C2-type Ig-SF structural domains [28].

The LILRB4 extracellular segment consists of two immunoglobulin-like structural domains, D1 and D2, each of which consists of antiparallel fragments; one contains three antiparallel fragments, and the other contains five antiparallel fragments. Compared to the other family members, the D1–D2 outer domain of LILRB4 adopts an unusually blunt interdomain angle of 107°, which is stabilized by hydrophobic interactions. In the D2 structural domain, LILRB4 shows two new 3_10_ helix regions, which are most closely related in sequence to the D4 structural domains of other LILRs.

The intracellular segment of LILRB4 is made up of three receptor tyrosine-based inhibitory groups (ITIMs). ITIM is commonly found on receptor molecules on the surface of some immune cells, and its primary role in signaling is to inhibit or negatively regulate immune responses. When the ITIM receptor binds to its ligand, the tyrosine in the ITIM region may be phosphorylated, triggering a series of signaling events that ultimately lead to an inhibitory response in immune cells [29].

ITIM are thought to be the only signaling motifs in LILRB4 [30]. LILRB4 contains three ITIMs with the sequences 335VTYAKY340, 387VTYAQL392 and 417SVYATL422, two SH3-binding motifs (PxxP) and proline-rich regions. LILRB4 transduces inhibitory signals by recruiting SHP-1/SHP-2 to its three ITIMs, leading to the termination of protein tyrosine kinase-mediated activation signals [31]. LILRB4 displays an internalized motif for tyrosine–x–x–valine/leucine in its tail [32], and the nature of these signals, which are involved in transactivation or inhibition, depends on the position of the different tyrosine residues in its tail ITIMs [31]. In the ITIM motif found in LILRB4, the tyrosines at positions 360, 412 and 442 are phosphorylation sites, of which Y412 and Y442 recruit SHP-2 to activate downstream signaling and further mediate the inhibition of T lymphocytes [30] (Figure 2).

LILRB4 has a tertiary structure, and two distinct surface patches in the D1 structural domain and the D1–D2 hinge region are highlighted. These characteristic structures result in it being unsuitable for binding MHC-I-like proteins, and they are also thought to aid LILRB4 in recognizing new ligands unrelated to β2-microglobulin [33].

Currently, there are various claims about LILRB4 ligands [31], although it has been established that integrin-αvβ3 is the ligand for gp49B, the mouse homologue of LILRB4. This suggests the emergence of a new ligand for an inhibitory immunoreceptor with a C2-type immunoglobulin-like structure [34]. However, it was found through experiments that a variety of integrin-αβ complexes are unable to activate human LILRB4 reporter cells [17], and it was therefore concluded that integrin-αβ complexes do not act as natural ligands for human LILRB4 molecules. In more recent findings, LILRB4 was shown to bind CD166 (activated leukocyte adhesion molecule, ALCAM) and serum apolipoprotein E (APOE), which mediate tumor cell growth and acute myeloid leukemia, respectively [35]. APOE specifically activates LILRB4, and both the N-terminal structural domain of APOE, P35, and W106 in the D1 structural domain of LILRB4 and Y121 in the linker region between the two structural domains D1–D2 are essential for APOE-mediated activation of LILRB4 [17]. Meanwhile, a recent study confirmed that human LILRB4 and its mouse homologue gp49B share a common ligand, fibronectin (FN) [36]. Human LILRB4 and mouse gp49B bind with submicromolar affinity to the N-terminal 30 kDa structural domain of human and mouse fibronectin (FN30), and the major LILRB4-binding sequence in FN30 is thought to be a conserved 20 aa Cys/Arg-rich sequence [37]. In the extracellular matrix, fibronectin is tethered by bound integrins; therefore, LILRB4 can be co-tethered to fibronectin (FN) with integrins in a cis-configuration on the same cell to form the LILRB4–FN–integrin triplex, which regulates downstream cellular activity [36].

The inhibitory receptor LILRB4 is mainly expressed on antigen-presenting cells (APC), including professional and nonprofessional antigen-presenting cells [38]. In specialized antigen-presenting cells, it is often expressed as a tolerance receptor on myeloid antigen-presenting cells such as dendritic cells, monocytes and macrophages [32] and plays an important role in the regulation of immune tolerance [33]. Published data on LILRB4 expression have also been obtained for non-specialized antigen-presenting cells such as endothelial cells [28]. LILRB4 expression on endothelial cells more significantly affects the immune tolerance effect of endothelial cells. Recently, data have shown that LILRB4 is highly expressed on myeloid-derived suppressor cells (MDSC), especially on monocyte-like myeloid-derived suppressor cells (M-MDSC) [39], and that LILRB4 orchestrates MDSC polarization, which has a role in inducing immune tolerance during the course of the disease [40]. Meanwhile, when immune cells such as B lymphocytes, T lymphocytes, and natural killer cells (NK cells) are activated, LILRB4 expression is upregulated in order to regulate cell differentiation or inactivation, thus exerting different immune effects [41]. LILRB4 expression has also been observed on other types of immune cells, such as innate immune cells resident in the central nervous system. It has been demonstrated that LILRB4 is upregulated in disease-associated microglia, suggesting that LILRB4 is a reliable surface marker for microglial activation [42].

## 3. LILRB4 and Immune Tolerance

Immune tolerance is a state in which the body, i.e., through T lymphocytes and B lymphocytes, does not respond to specific antigens that are supposed to elicit specific responses to antigens. Thus, these lymphocytes are not activated in response to antigenic stimulation, they do not produce specific immune effector cells and specific antibodies and they are therefore unable to facilitate the procession of a normal immune response [43,44]. Immune tolerance can be divided into categories of central or peripheral immune tolerance, depending on where it occurs [45,46,47]. Central immune tolerance is established during the development of T and B cells in the thymus and bone marrow. It is achieved through a mechanism of negative selection of lymphocytes against self-antigens, in which the majority of cells in the body that are self-reactive are removed in the process [48,49]. However, approximately 25–40% of autoreactive T cells and approximately 40% of autoreactive B cells escape from the central tolerance process. Therefore, these escaped autoreactive cells are dealt with to maintain the body’s immune tolerance through the action of peripheral tolerance mechanisms, including clonal inactivation, clonal deletion and via regulatory T cells (Treg) [50,51,52,53].

The induction, maintenance and termination of immune tolerance are associated with the onset, progression and regression of numerous clinical diseases [54]. In clinical practice, for autoimmune diseases and transplantation immune tolerance, physiological tolerance to self-antigens is often re-established through the use of immunosuppressants and other means [55,56,57]. Meanwhile, for chronic infections and tumors, methods are needed in order to break the pathologic tolerance, restore the normal immune response and ultimately achieve the purpose of removing pathogens and killing tumor cells. Thus, in the complex biology of immune system regulation, the mechanisms that down-regulate host immune responses are as important as those that activate them [58]. Studies have shown that the LILRB family of immune checkpoint proteins has great potential in inducing immune tolerance, and LILRBs have emerged as a topic that deserves in-depth study [59].

Numerous studies have been published in recent years on how the LILRB family members play an important role in inducing immune tolerance, suggesting that the LILRB subfamily has potential as an attractive therapeutic target for the next generation of immunotherapies. The LILRB subfamily is usually overexpressed in cells associated with immunosuppression, such as immunosuppressive M2-type macrophages and tolerogenic dendritic cells [60,61]. The upregulated expression of LILRB1 tolerizes dendritic cells, hinders adaptive immunity and promotes immune evasion [62,63]. LILRB1 and LILRB2 compete with CD8^+^ T cells to bind HLA class I molecules and inhibit antigen-presenting cell activation, resulting in an inhibitory effect on downstream T cell activation and proliferation. Moreover, the interaction of LILRB1 and LILRB2 with human leukocyte antigen-G (HLA-G) provides a potential immune escape mechanism for tumors and fetuses to overcome immune surveillance and avoid immune attack [64,65]. In addition, expression of the inhibitory receptor LILRB3 on human myeloid cells is associated with the upregulation of immunosuppressive genes on immunosuppressive M2 macrophages [66].Expression of LILRB4 and LILRB5 in macrophages modulates the JAK/STAT signaling pathway and mediates the upregulation of immunosuppressive cytokines, such as interleukin 10 (IL-10), as well as the downregulation of inflammatory cytokines, such as interleukin 8 (IL-8) [67], which negatively modulates the immune response of the organism.

Similarly, LILRB4, as a member of the LILRB family of inhibitory receptors, is particularly important in the body’s immune response by exerting inhibitory effects and inducing immune tolerance. LILRB4 contains inhibitory ITIM motifs in its cytoplasmic structural domain and mediates the inhibition of cellular activation through SHP-1/SHP-2 recruitment to reduce the activation of the body’s immune response occurrence [17]. Moreover, the resulting induced differentiation of suppressor cells, such as CD8^+^ suppressor T cells (Ts cells), causes them to proliferate in large numbers and can induce CD4^+^ T helper cell (Th cell) dysfunction in an HLA-restricted manner, as shown in tissue culture [60]. In this way, uncontrolled lymphocyte proliferation following exposure to antigenic stimuli can be avoided, and the generation of aberrant immune responses that target host antigens and lead to the development of autoimmune diseases can be prevented [58]. Thus, activation of the inhibitory receptor LILRB4 promotes tolerance and immunosuppression, which could play an important role in the induction of immune tolerance in the body and is expected to be an effective therapeutic target in different disease settings.

## 4. Role of LILRB4 in the Immune Response

### 4.1. High Expression of LILRB4 Decreases Cytokine Expression and Induces Inhibitory Cell Differentiation

In certain disease states, such as in tumors or in the presence of certain cytokines (e.g., IL-10, IFN-β, etc.), elevated expression levels of LILRB4 on APCs are induced. LILRB4 contains inhibitory ITIM motifs in its cytoplasmic structural domains that mediate the inhibition of cellular activation through the recruitment of regulatory tyrosine phosphatases [60]. Elevated LILRB4 induces Th cell exhaustion and inhibits differentiation of IFN-γ-producing cytotoxic T lymphocytes (CTL). LILRB4 not only inhibits the immune effector activity of T lymphocytes but also exerts an inhibitory effect on NK cells. Fibronectin (FN) is thought to be a key component of the extracellular matrix that is prevalent in the tumor microenvironment, aiding in tumor metastasis and immune evasion. The binding of FN to integrin receptors transmits activation signals to NK cells, causing them to exert killing effects. Activated NK cells upregulate the expression of LILRB4, which is a co-inhibitory receptor for fibronectin receptor integrins, blocking activation signals and transmitting inhibitory signals to NK cells, resulting in a reduction in the killing effect of NK cells [68].

High expression of LILRB4 inhibits the production of intracellular co-stimulatory molecules and inflammatory cytokines (e.g., IL-6, IL-1β, TNF-α, etc.) and decreases the transcription of inflammatory exosomal microRNA [69,70]. At this point, APCs such as dendritic cells and monocytes become tolerant in the presence of LILRB4. When cells highly expressing LILRB4 are induced to become tolerant, they will further exert a tolerogenic effect and facilitate an immune tolerance cascade response. Tolerant DCs reflect a state of cellular suppression, wherein DCs are unable to accomplish the induction and maintenance of Th cell activation and maturation, and it has been shown that DCs characterized by tolerance are inducers of two cell populations, regulatory T cells (Treg cells) and Ts cells [71]. DC cell–T cell interactions are dependent on TCR–MHC recognition and are bidirectional. Thus, high expression of LILRB4 induces DC cells to become tolerant. Similarly, high expression of LILRB4 on tolerant DCs induces the generation of Treg cells and Ts cells in the naive T cell population, which is mainly generated by the induced increase in the expression of BCL6 on CD8^+^ T cells, leading to the differentiation of CD8^+^ T cells to Ts cells [72]. Differentiated Treg cells and Ts cells can in turn direct the differentiation of immature DCs into tolerant DCs by inducing the expression of the inhibitory receptor LILRB4, thus forming a complete immune tolerance cascade response.

### 4.2. LILRB4 Modulates Intracellular Signaling Pathways

When highly expressed on antigen-presenting cells, LILRB4 influences cellular signaling pathways through its ITIM inhibitory motifs. LILRB4 signaling involves SHP-1/SHP-2, which is dependent on tyrosine phosphorylation of the ITIM motif and the intact SHP-1 carboxyl SH2 structural domain [73]. Elevated levels of LILRB4 transactivation result in reduced phosphorylation levels of intracellular transforming growth factor (TGF)-activated kinase 1 (TAK1), nuclear factor kappa-B (NF-κB), and mitogen-activated protein kinases (MAPKs), leading to a weakened signaling response to NF-κB ligand receptor activator (RANKL) [74]. Additionally, LILRB4 on macrophages activates JAK/STAT signaling pathway genes, influencing cytokine expression and contributing to the regulation of immune tolerance in macrophages [75] (Figure 3).

### 4.3. LILRB4 Plays a Unique Role in Epigenetic Modification

It has been reported that LILRB4 coordinates MDSC polarization by inducing the differentiation of M-MDSCs toward a tumor-promoting M2 phenotype, and that high expression of LILRB4 decreases the expression of anti-tumor miRNAs in MDSCs, including miR-1a-3p, miR-133a-3p and miR-206-3p [40]. Fat mass and obesity-associated protein (FTO) is an enzyme involved in removing a methyl group from RNA, specifically N6-methyladenosine (m6A). Scientists are interested in FTO due to its role in RNA demethylation, impacting various biological processes related to gene expression and metabolism [76]. Additionally, FTO has been identified as having an oncogenic role in multiple cancers, and it targets LILRB4 in acute myeloid leukemia (AML) cells. Blocking FTO using drugs significantly reduces the self-renewal of leukemic stem cells and alters the immune response by primarily suppressing expression of the immune checkpoint gene LILRB4 [77].

## 5. Regulatory Role of LILRB4 Molecules in Disease

### 5.1. LILRB4 and Autoimmune Tolerance

Systemic lupus erythematosus (SLE) is a chronic diffuse connective tissue disease of unknown etiology that can invade multiple systems throughout the body. The afflicted body produces large amounts of autoantibody IgG, which cause the immune system to attack its own tissues, resulting in damage to multiple organs and tissues throughout the body [78]. Clinical treatment of SLE is often achieved through the use of rituximab to target and remove virtually all B cells, both initial and activated, with predictable effects on the immune system. Therefore, pinpointing autoantibody-producing plasmablasts (PBs) and plasma cells (PCs) by finding effective target molecules is essential for downregulating pathogenic plasma cell activity. It was found that PBs and PCs from untreated SLE patients would have high LILRB4 expression, and that the level of LILRB4 expression was positively correlated with the production of anti-double-stranded DNA IgG in serum [19,79]. Therefore, utilizing the highly expressed LILRB4 molecule to recognize pathogenic PBs/PCs and target pathogenic PBs/PCs could provide a new avenue for effective treatment of SLE.

Fibronectin (FN) is a major macromolecule in the extracellular matrix (ECM) and a physiological ligand for LILRB4. In BXSB/Yaa mice, a well-established SLE model mouse, the interaction of FN with LILRB4, which is elevated on the surface of plasma cells, promotes antibody class switching in plasma cells to produce more pathogenic autoantibodies. It was shown that blocking the interaction between LILRB4 and FN using recombinant proteins of LILRB4 or monoclonal antibodies to LILRB4 significantly reduces the increase in pathogenic autoantibody IgG and increases the amount of protective IgM antibodies, which can be used for the treatment of SLE in BXSB/Yaa mice [37]. These findings suggest that LILRB4 can be used both as a marker of SLE based on its high expression on pathogenic plasma cells and as an emerging target molecule for the treatment of SLE. However, how to effectively translate these findings from animal models of SLE into clinical roles is still an urgent question that needs to be addressed.

Multiple sclerosis (MS) is a chronic autoimmune, inflammatory and neurodegenerative disease affecting the central nervous system (CNS). Multiple sclerosis is characterized by immune dysregulation, leading to the infiltration of immune cells into the central nervous system and triggering demyelination, axonal damage and neurodegeneration [80,81]. Jensen et al. showed that the expression levels of LILRB4 in circulating monocytes of patients is significantly reduced during MS episodes. A reduction in the level of LILRB4 expression leads to a drastic increase in proinflammatory cytokines in patients’ bodies, causing neuroinflammation and leading to serious consequences such as limb paralysis [82]. In this regard, it has been demonstrated that MS disease progression can be inhibited by the use of IFN-β and vitamin D, which induces increased levels of LILRB4 on APCs during MS exacerbations [83]. Therefore, the co-induction of LILRB4 elevation in vivo, through the use of IFN-β and vitamin D, is expected to be a new target for intervention in relapsing multiple sclerosis [84].

Inflammatory bowel disease (IBD), which includes Crohn’s disease and ulcerative colitis, is a chronic, recurrent autoimmune disease. It has a prolonged course, is recurrent and remains incurable [85,86]. The severity of IBD correlates with the activation of MAPK and NF-κB as well as the production of proinflammatory cytokines (e.g., IL-6, IL-1β, TNF-α, etc.) in patient macrophages, it was shown that LILRB4, via SHP-1, negatively regulates the activation of the MAPK and NK-κB pathways as well as the production of proinflammatory cytokines during IBD episodes, thus controlling the development of IBD. Thus, LILRB4 functions as a negative regulator of macrophage responses to pathogenic bacteria and chronic intestinal inflammation, and it has the potential to be a target for IBD disease therapy [87] (Figure 4).

### 5.2. LILRB4 and Transplantation Immune Tolerance

Research on inducing immune tolerance in transplantation has long focused on inducing recipient-specific immune tolerance to the transplanted organ, i.e., the allograft, without the need for lifelong treatment that usually involves toxic immunosuppressive drugs. The search for therapies that may induce specific immune tolerance is ideally conducted through short-term interventions that target only pathogenic immune responses without compromising protective host immune responses [58]. Recent studies have shown that LILRB4, through its potent in vivo inhibitory effects, is expected to be a key protein molecule in inducing the body to develop transplantation immune tolerance after transplantation. The LILRB4 protein is thus expected to be an attractive target for inducing the development of specific immune tolerance in the body for treatment in cases of transplantation rejection.

Nonspecialized antigen-presenting cells (mainly endothelial cells) represent the first barrier between the donor organ and the recipient’s immune system. The induction of immune tolerance in endothelial cells is a therapeutic strategy to induce tolerance of recipients to allogeneic grafts without compromising their immune system, in which the upregulation of LILRB4 on endothelial cells plays an important role [28]. The endothelial cells of grafts play a key role in inducing rejection and in activating T cells to exert immune effects through direct or indirect allorecognition pathways [88]. However, induced LILRB4 upregulation on endothelial cells via allogeneic antigen-specific Ts cells in the blood circulation of transplant recipients results in the decreased production of co-stimulatory molecules and adhesion molecules, which can lead to tolerance and a reduction in the development of graft rejection in endothelial cells. LILRB4 precursor RNA is conserved in the nucleus of resting endothelial cells, and processing of LILRB4 precursor mRNA is triggered when Ts cells interact with endothelial cells or when endothelial cells are exposed to IL-10 and IFN-α. The production of processed, mature LILRB4 transcripts is accompanied by the production of LILRB4 proteins, which can thus play an important role in inducing immune tolerance to transplantation [89]. Modulation of inhibitory receptor LILRB4 expression on endothelial cells facilitates the first line of defense in inducing graft tolerance, which is essential for the long-term survival of allograft organs [90].

Kidney transplantation is the first and most numerous and technically mature of the types of large organ transplantation in China. Immunosuppressive regimens initiated for renal transplant recipients, such as the administration of immunosuppressive drugs such as tacrolimus, mycophenolate mofetil and sirolimus, have been found to induce a rise in the expression of LILRB4 on antigen-presenting cells, thereby inducing cellular tolerance, which facilitates better allograft survival [91]. Meanwhile, the mTOR inhibitor rapamycin is the most commonly used immunosuppressive therapy after renal transplantation [92], and it has been shown that renal transplant recipients suffering from chronic graft nephropathy experience a significant increase in LILRB4 expression in renal biopsy tissues after prolonged rapamycin treatment. This directly induces an increase in the population number of suppressive cells, such as Treg cells, as well as an expansion of the CD8^+^CD28^−^ Ts cell population [93]. Interestingly, the use of vitamin D3, IL-10 and IFN-α, among others, can also upregulate LILRB4 expression, thereby assisting in the maintenance of tolerance after transplantation [94,95]. Therefore, inducing the upregulation of LILRB4 expression can be a good way to induce the onset of organismal immune tolerance, thereby reducing the probability of post-transplantation rejection.

In cardiac transplantation studies, high levels of LILRB4 expression on circulating Ts cell-induced APCs in peripheral blood were associated with a significant reduction in rejection [60].Transplanting splenocytes from heterozygous C57 LILRB4(-/-) mice into BALB/C mice did not cause severe graft-versus-host disease, indicating the crucial role of LILRB4 in inducing immune tolerance post-transplantation [96]. The soluble LILRB4-Fc recombinant protein exhibits robust immunosuppressive effects, leading to the exhaustion of CD4+ Th cells and inhibiting the differentiation of IFN-γ-producing CD8+ CTL cells [97,98]. Furthermore, CD8+ T cells treated with LILRB4-Fc can differentiate into regulatory T cells (Ts cells) and acquire suppressive functions, as evidenced by the significant upregulation of BCL6. In a humanized NOD/SCID mouse model, LILRB4-Fc induces tolerance to allogeneic human islets and reverses rejection reactions [99]. Given its potent immunosuppressive effects, LILRB4-Fc is considered to be a potential approach for treating transplant rejection and inducing immune tolerance post-transplantation [100].

### 5.3. LILRB4 and Maternal–Fetal Immune Tolerance

LILRB4 is a central inhibitory receptor for uterine dendritic cells (uDCs) and decidual myeloid-derived suppressor cells (dMDSCs), and it plays an important immunomodulatory role at the maternal–fetal interface, thereby ensuring that the process of pregnancy proceeds normally [38,101]. HLA-G is a non-classical MHC class I molecule that is selectively and highly expressed in extrachorionic trophoblast cells invading the uterine metaphysis [102]. The expression of HLA-G has an important role in maternal–fetal immune tolerance and the maintenance of normal pregnancies. This role is correlated with the expression of the inhibitory receptor LILRB4. Prior to the occurrence of an immune response, i.e., in the absence of antigenic stimulation, HLA-G upregulates LILRB4 expression on cells such as antigen-presenting cells, NK cells, etc., inducing maternal–fetal tolerance to ensure that a normal pregnancy occurs [103,104]. Meanwhile, increased maternal vitamin D intake during pregnancy helps to increase the expression of LILRB4 mRNA in umbilical cord blood, which induces the differentiation of dendritic cells, macrophages, and other cells into tolerogenic cells. Additionally, there is early induction of a tolerogenic immune response [105].

Toxoplasma gondii infection is an infectious disease of eukaryotic intracellular parasites commonly seen during pregnancy. Toxoplasma gondii infection may cause retinopathy and can lead to life-threatening disseminated infections involving the central nervous system [106,107]. If Toxoplasma gondii infection occurs during pregnancy, a number of complications may arise, including stillbirth, miscarriage and congenital malformations. When Toxoplasma gondii infection occurs during pregnancy, the dendritic cells, by inducing upregulation of LILRB4 on both uterine and decidual macrophages, can cause changes in the expression of cell membrane molecules (CD80, CD86, HLA-DR or MHC II), the synthesis of arginine metabolizing enzymes and the secretion of cytokines to suppress the likelihood of an abnormal pregnancy by impairing the activation function of M1-type macrophages and enhancing the tolerance of M2-type macrophages. Additionally, uDCs may provide a potential approach for the treatment and prevention of congenital toxoplasmosis [108,109] (Figure 5).

### 5.4. Studies on the Clinical Application of Blocking LILRB4 Receptors in Tumor Immunotherapy

As a member of the immune checkpoint molecules, the role of LILRB4 in tumors is generally to promote tumor cell invasion and migration and to inhibit the proliferation and activation of immune cells such as T lymphocytes [110]. LILRB4 is a well-known marker of monocytic leukemia, and it supports tumor cell infiltration into tissues and inhibits the proliferation and activation of T lymphocytes through a signaling pathway involving APOE–LILRB4–SHP2–uPAR–ARG1 in AML cells [17]. In non-small cell lung cancer (NSCLC), LILRB4 enrichment in tumor cells is often predictive of advanced disease progression and poorer overall survival [111]. In NSCLC, LILRB4 recruits SHP-2 and SHP-1, which results in subsequent ERK1/2 signaling activation, mediating the epithelial–mesenchymal transition (EMT) and increasing vascular endothelial growth factor (VEGF-A) expression in NSCLC cells in support of tumor cell invasion and angiogenesis [112]. This high expression of LILRB4 leads to an increased incidence of recurrence and poor prognosis in NSCLC patients with resected tumors [113]. However, there are exceptions, suggesting that LILRB4 may also play a tumor-suppressive role in tumor development, such as in chronic lymphocytic leukemia (CLL), where LILRB4 is ectopically expressed and a selective marker for CLL malignancy. Although LILRB4 expression inhibits Akt kinase activation upon B-cell receptor (BCR) stimulation, it can functionally contribute to the regulatory network controlling tumor progression by inhibiting the Akt pathway [114].

In a variety of tumors that target LILRB4 as a protein, such as AML, a commonly used immunotherapeutic strategy is to use a monoclonal antibody or inhibitor against LILRB4 to inhibit the attachment of LILRB4 to its ligands, such as APOE/FN, to block LILRB4 signaling, inhibit tumor cell invasion and migration and restore the immune effector activity of immune cells. The attachment and interaction of FN with LILRB4 represents a “matrix checkpoint” through which the extracellular matrix inhibits myeloid cells. By blocking this interaction, tumor-associated myeloid cells may acquire a stimulatory phenotype that elicits an increase in anti-tumor T cell responses [115]. In contrast, disruption of the LILRB4–APOE interaction by a humanized antibody against LILRB4 also reverses T cell suppression and results in potent activity in blocking the development of monocytic AML. A study on the identification and development of a LILRB4-specific human monoclonal antibody, h128-3, demonstrated its potent activity in blocking AML development in monocytes in a variety of mouse models, including patient-derived xenograft mice and homozygous immunocompetent AML mice. This specific human monoclonal antibody h128-3 blocks the inhibitory signaling that would normally occur upon LILRB4 binding to APOE by binding to LILRB4, resulting in effective treatment of AML by reversing the multiple effects of T-cell suppression, inhibiting tissue infiltration of AML cells by monocytes and attenuating antibody-dependent cytotoxicity and antibody-dependent cellular phagocytosis. Similarly, it has been shown that LILRB4 expression is significantly elevated and supports tumor cell migration and invasion in the tumor-associated macrophages (TAMs) of many solid tumors. However, after treatment with an anti-LILRB4 antibody, mice show a significant reduction in tumor load and improved survival, suggesting that LILRB4 strongly suppresses tumor immunity in the tumor microenvironment and that this suppression can be attenuated by administering a monoclonal antibody against LILRB4 to provide effective antitumor therapy. Therefore, targeting LILRB4 with monoclonal antibodies is an effective therapeutic strategy for the treatment of tumor diseases [116].

### 5.5. Studies on the Association of LILRB4 with Other Diseases

In viral infections, LILRB4 usually acts positively to control viral infections. It has been reported that LILRB4 expression is increased during acute infections with lymphocytic choroid plexus meningitis virus (LCMV), influenza virus and acute respiratory syndrome coronavirus 2 (SARS-CoV-2) [117,118,119]. Its expression is correlated with a strong antibody response in influenza patients as well as with the proliferation of effector CD8^+^ T cells in mice [118,120]. An increase in LILRB4 in activated NK cells during cytomegalovirus infection has also been reported [121]. Studies have shown that acute encephalitis caused by a number of viruses, including the neurologic Zika virus ZIKV, leads to an increase in the number of cells in the brain that express LILRB4, which is essential for controlling the infection [41]. LILRB4 controls viral load in viral infections by ensuring normal cell maturation and function of NK cells, resulting in effective viral clearance and patient survival. Plasma monocytes, a class of cells with high LILRB4 expression, are often found in and around the high endothelial veins of inflamed lymph nodes. They express CD62L and CXCR3 and produce large amounts of type I interferons upon stimulation by influenza virus or CD40L [102], whereas type I interferons often induce upregulation of LILRB4 expression early in infection, which is essential for inhibiting viral proliferation [83].

LILRB4 has also been implicated in the pathological process of various inflammatory diseases, where it significantly inhibits the production of TNF-α, a key proinflammatory cytokine induced by Fcγ-RI (CD64), thereby modulating the development of inflammation [122]. Atherosclerosis is a chronic inflammatory disease, and LILRB4 expression is upregulated in atherosclerotic lesion sites in human coronary arteries and mouse aortic roots. A lack of LILRB4 markedly accelerates the development of atherosclerotic lesions and increases plaque instability, as evidenced by an increase in lipid infiltration and a decrease in the number of collagenous components and smooth muscle cells. The proinflammatory effects of LILRB4 deficiency are mediated in part by increased NF-κB signaling activation due to decreased SHP-1 phosphorylation [123]. 

Nonalcoholic fatty liver disease (NAFLD) is an increasingly prevalent liver condition characterized by hepatic steatosis, usually accompanied by systemic inflammation and metabolic disturbances. LILRB4 is a regulator of NAFLD, and LILRB4 recruits SHP-1 to inhibit TNF receptor-associated factor 6 (TRAF6) ubiquitination, which in turn inhibits cascade inactivation of NF-κB and mitogen-activated protein kinase. In a mouse model of nonalcoholic fatty liver disease, overexpression of LILRB4 largely reversed intrinsic hepatic steatosis, inflammation and metabolic disturbances. Improving the expression or activation of hepatic LILRB4 through its targeting is a promising therapeutic strategy against NAFLD and related hepatic and metabolic diseases [124]. 

Chronic obstructive pulmonary disease, or chronic bronchitis, is a common lung disease that causes restricted airflow and breathing problems. LILRB4 was found to be upregulated on interstitial macrophages in human chronic obstructive pulmonary disease and mouse emphysema models. Upregulation of LILRB4 may exert a protective effect against emphysema formation by decreasing the expression of matrix metalloproteinase MMP-12 in the lungs [125]. Moreover, in other inflammatory lung conditions, such as acute lung injury, a lack of LILRB4 expression would exacerbate acute lung injury through NF-κB signaling in bone marrow-derived macrophages [126].

As the resident innate immune cells of the CNS, microglia play important roles in both physiological and pathological conditions. LILRB4 is a reliable surface marker for activated microglia, and microglial TGF-β signaling is involved in the regulation of LILRB4 expression during lipopolysaccharide (LPS)-induced microglia activation [42]. LILRB4 expression is upregulated in the brain in the microglia of aged mice as well as around plaques in a mouse model of Alzheimer’s disease, and its expression has been associated with immunosuppressive and immune tolerance functions [127,128,129].

It has been found that LILRB4 is constitutively expressed on the surface of mast cells and that it counter-regulates mast cell activation mediated by immunoglobulins produced in response to adaptive immunity in vivo [130]. LILRB4 inhibits IgE-dependent mast cell activation in vitro by recruiting src homology structural domain type 2-containing SHP-1 to the cell membrane via its three tyrosine-based inhibitory motifs (ITIMs). LILRB4-deficient mice exhibit higher IgE levels and increased incidence and severity of mast cell-dependent allergic inflammation compared to LILRB4-expressing mice [131,132]. LILRB4-deficient mice also have a significantly higher and faster mortality rate in active systemic anaphylaxis [133,134]. Observations regarding the in vivo role exertion of LILRB4 in two experimental models of allergic inflammation and infection with Ascaris lumbricoides larvae revealed that LILRB4 acts as an inhibitor of allergic inflammatory responses on eosinophils [135]. Thus, LILRB4 inhibits allergic responses driven by adaptive immune responses in vivo. Allergic diseases are caused not only by the hyperactivation of cells but also by the lack of receptors that inhibit these activated responses [136,137,138]. Thus, LILRB4 has potential as a promising therapeutic target against allergic diseases.

Cardiac hypertrophy is an adaptive response to physiological and pathological overload, but sustained pathological overload induces maladaptation and cardiac remodeling, leading to heart failure [139,140]. LILRB4 was found to be expressed in both myocardial tissue and cultured cardiomyocytes under normal conditions, but it was significantly reduced in mouse hearts after aortic banding and in angiotensin II-treated cardiomyocytes. Overexpression of LILRB4 in the heart attenuates cardiac hypertrophy, fibrosis and dysfunction in response to pressure overload, and it inhibits angiotensin II-induced cardiomyocyte hypertrophy in vitro. It has been shown that LILRB4 prevents pathological cardiac hypertrophy through SHP-2-dependent inhibition of the NF-κB pathway, and it may serve as a potential therapeutic target for cardiac hypertrophy [141].

## 6. Conclusions

In this review, we summarized information on LILRB4 related to its structure, ligand molecules and expression in cells, highlighting how its inhibitory role in immune processes induces immune tolerance. We also discussed how it plays a unique role in different diseases to regulate the disease process. LILRB4 is a type of inhibitory receptor that is mainly expressed on antigen-presenting cells and is upregulated on dendritic cells, macrophages and other immune cells stimulated by Ts cells or some cytokines, thus inducing the differentiation of these cells into tolerant cells. These, in turn, induce the differentiation of inhibitory Ts cells and Treg cells and induce the dysfunction of immune effector cells such as Th cells and CTL cells in order to produce the immune-tolerance cascade reaction, thereby exerting an immune-suppressing effect.

In cells, LILRB4 can recruit tyrosine phosphatase SHP-1/SHP-2 via the ITIM motif, inhibiting the production of proinflammatory cytokines by negatively regulating the activation of signal pathways such as NF-κb and MAPK. This results in the significant alleviation of autoimmune diseases. The results show that LILRB4 and its derived recombinant protein LILRB4-Fc have strong immunosuppressive effects and play a key role in the treatment of autoimmune diseases, the induction of transplantation tolerance and the induction of maternal–fetal immune tolerance. This can be used as part of an immunotherapeutic strategy in the follow-up treatment of these clinical contexts.

In the treatment of tumors, a commonly adopted immunotherapy strategy is to block the signaling of LILRB4 by inhibiting its association with its ligands such as APOE/FN using mAbs or inhibitors against LILRB4, thereby preventing the invasion and migration of tumor cells and restoring the immune effector activity of immune cells. 

We also summarized the role of LILRB4 in numerous disease processes, such as viral infection and inflammatory diseases, which suggests that LILRB4 may be a relevant target molecule in many kinds of diseases, signifying its broad research prospects. Although a number of application directions for LILRB4 have been identified, attention is also needed to address some related questions: How can findings in animal models be applied in clinics for effective treatment? How can LILRB4 be better targeted to achieve therapeutic effects without interfering with other normal immune responses in the body? These are some of the directions that need to be considered in the future study of LILRB4.

## Figures and Tables

**Figure 1 biomolecules-14-00187-f001:**
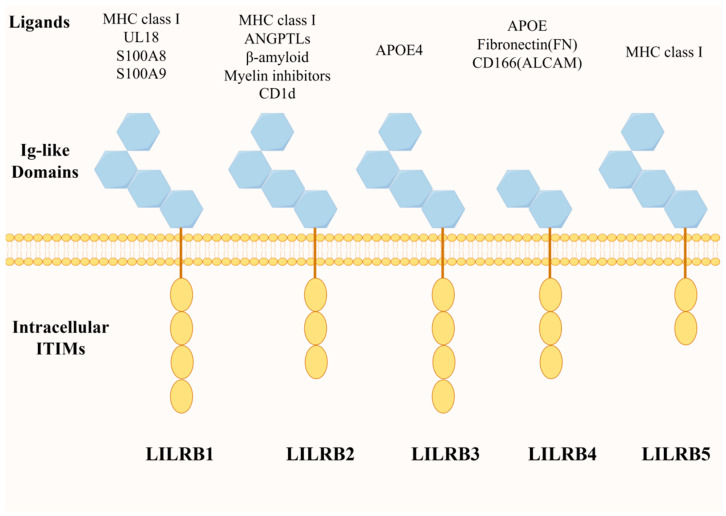
LILRB family member structures and their ligand molecules. The LILRB family consists of five members: LILRB1, LILRB2, LILRB3, LILRB4 and LILRB5; all of them are mainly composed of intracellular ITIM motifs and extracellular Ig-like structural domains, but there are subtle differences in composition (illustrated using www.figdraw.com, accessed on 25 January 2024).

**Figure 2 biomolecules-14-00187-f002:**
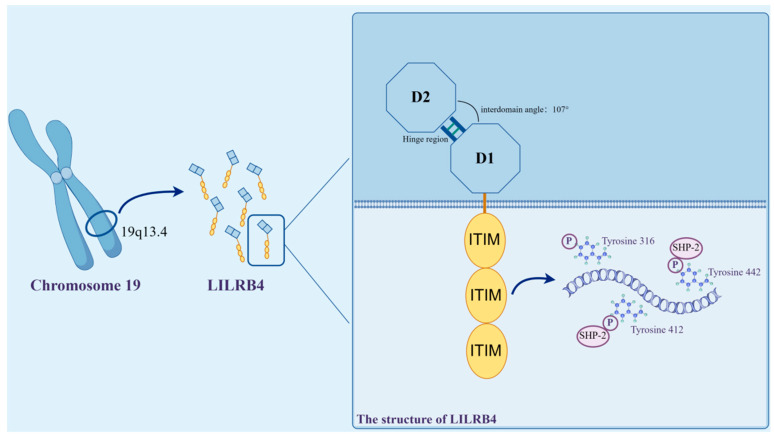
Structure of LILRB4. The LILRB4 gene is localized to the leukocyte receptor cluster on chromosome 19q13.4. The extracellular segment consists of two immunoglobulin-like structural domains, D1 and D2, whose D1–D2 ectodomain employs an unusually blunted 107° interdomain angle for its stabilization via hydrophobic interactions. The intracellular segment, on the other hand, is comprised of three immunoreceptor tyrosine-based inhibitory motifs (ITIM), and within the constituent ITIM motifs, the tyrosines at positions 360, 412 and 442 are phosphorylation sites. Of these, Y412 and Y442 recruit SHP-2 to activate downstream signaling and further exert relevant immune effects (illustrated using www.figdraw.com, accessed on 25 January 2024).

**Figure 3 biomolecules-14-00187-f003:**
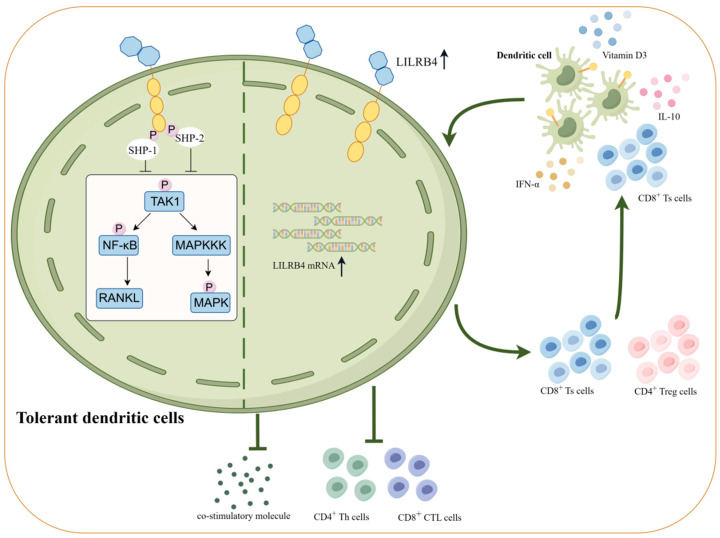
Immune tolerance cascade induced by elevated LILRB4 expression on antigen-presenting cells. LILRB4 precursor RNA is retained in the nucleus of dendritic cells, and treatment of LILRB4 precursor mRNA is triggered when dendritic cells are exposed to IL-10, IFN-α, VD3 and Ts cells. The production of mature LILRB4 transcripts after treatment is accompanied by an elevated expression of LILRB4 protein. LILRB4, through tyrosine phosphorylation of its intracellular ITIM motifs, signals for the recruitment of SHP-1/SHP-2, leading to lower phosphorylation levels of TAK1/NF-κB/MAPK and further reducing RANKL-induced signaling. Moreover, high expression of LILRB4 inhibits the transcription of intracellular co-stimulatory molecules, inflammatory cytokines, and inflammatory exosomal microRNAs. Under the multiple effects of LILRB4, dendritic cells become tolerogenic, induce the differentiation of Treg and Ts cells and inactivate CD4^+^Th cells and CD8^+^CTL cells. The differentiated Treg and Ts cells can in turn direct the differentiation of immature DCs to tolerogenic DCs by inducing the expression of the inhibitory receptor LILRB4, thus forming a complete immune tolerance cascade (illustrated using www.figdraw.com, accessed on 25 January 2024).

**Figure 4 biomolecules-14-00187-f004:**
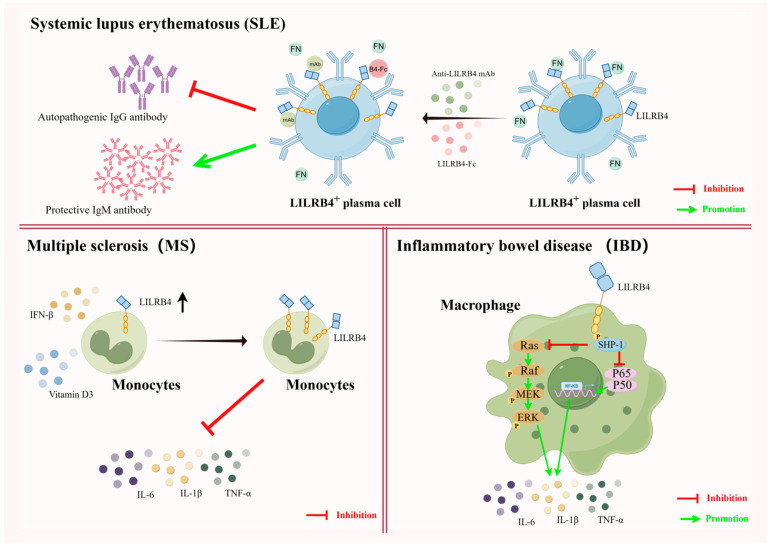
Diverse uses of LILRB4 in autoimmune diseases. In systemic lupus erythematosus (SLE), the plasma cells of SLE patients express high levels of LILRB4. Fibronectin (FN) has the property pathogenicity when it binds to LILRB4, present at high levels, in plasma cells. When FN binding to LILRB4 is blocked by a monoclonal antibody against LILRB4 or the recombinant protein of LILRB4, the increase in the levels of pathogenic autoantibody IgG is significantly inhibited, and the amount of protective antibody IgM is increased, thereby alleviating the disease. In both multiple sclerosis and inflammatory bowel disease (IBD), LILRB4 exhibits roles that are distinct from those in SLE. In MS, increased levels of LILRB4 on APCs can be induced by the use of IFN-β and vitamin D, which can reduce the production of proinflammatory cytokines and effectively suppress disease progression. In IBD, LILRB4, which is expressed on macrophages, can reduce the activation of proinflammatory cytokines by negatively regulating the activation of ERK and NF-κB signaling pathways, and it may also alleviate disease (illustrated using www.figdraw.com, accessed on 25 January 2024).

**Figure 5 biomolecules-14-00187-f005:**
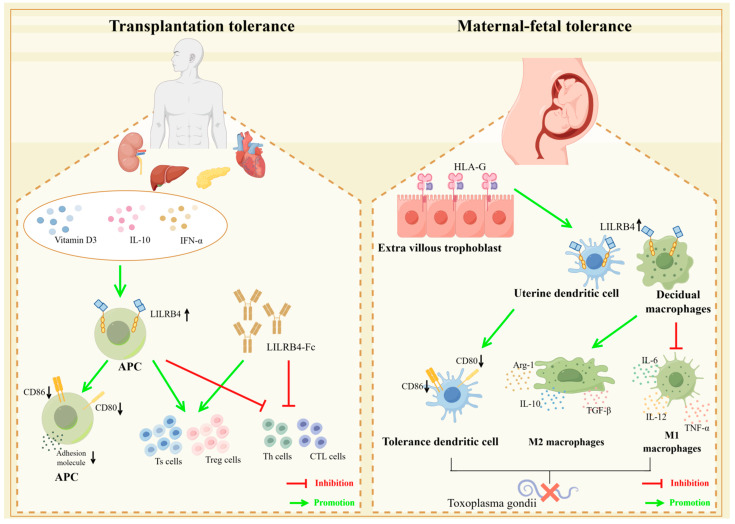
LILRB4 plays a role in transplant tolerance and maternal–fetal tolerance. In transplantation tolerance induction, upregulation of LILRB4 expression on APCs using vitamin D3, IL-10 and IFN-α or the use of recombinant LILRB4 protein induces a decrease in co-stimulatory molecules such as CD80/CD86 on APCs. It also induces Th and CTL cell dysfunction and causes expansion of Ts and Treg cells, which could be effective in inducing transplantation tolerance. This can effectively induce the generation of graft tolerance. During maternal–fetal tolerance, HLA-G is highly expressed in extrachorionic trophoblast cells, which induces upregulation of LILRB4 on uterine dendritic cells and metaplastic macrophages and induces the generation of tolerant DCs by altering the expression of cell membrane molecules such as CD80, CD86, etc. At the same time, it enhances M2-type macrophage differentiation and attenuates M1-type macrophage proliferation, through which the creation of tolerant environments could potentially lead to a pathway for congenital Toxoplasma gondii treatment and prophylaxis (illustrated using www.figdraw.com, accessed on 25 January 2024).

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
