# Peer review of "LILRB4 Checkpoint for Immunotherapy: Structure, Mechanism and Disease Targets"

_biomolecules, 2024, doi:10.3390/biom14020187_

Round 1
Reviewer 1 Report
Comments and Suggestions for Authors
The manuscript titled “LILRB4 checkpoint for immunotherapy: Structure, Mechanism and Disease targets”, submitted by Dr Yizhou Zou and his group, is an interesting review of current relevance that attempts to describe the most relevant findings on the role of LILRB4 in diseases; as autoimmune and oncological diseases, as well as its possible use as a therapeutic alternative. Although the specific ligand for LILRB4 is not known and knowledge of the role of LILRB4 is limited, this review describes the advances to date in the knowledge of this important protein. However, there are details that must be addressed and described below to be clearer for the reader.
-There are no references in the first part of the introduction
- Figure abut graphical representation of the LILR family and its domains is suggested.
- In general, there are sections where references are needed, such as in the paragraph on lines 152-161
- The first part of section 2 of LILRB4 and immune tolerance lines 152-180 is too extensive in terms of central and peripheral tolerance. It is suggested to go more directly to the role of LILRB4.
- There are serious errors in the nomenclatures of the cell types of the immune system and the descriptions pf abbreviations are repetitive. (example; Ts, NK, CD8+ Ts)
- The role of LILRB4 in autoimmune diseases such as SLE and IBD is unclear. Describe in greater detail the findings of the role of LILRB4 in these diseases and what is represented in Figure 3 is not very clear according to what is described in the section.
- Figure from section 4.2 is suggested, to help the reader and be clearer in the mechanisms described in the section. The same for the other sections (4.3, 4.4 and 4.5) described.
Author Response
Dear Reviewer:
We are very grateful to your valuable comments on this article, which have been helpful in improving the quality of the manuscript. We have carefully considered and responded to each of the suggestions made by the reviewer. The following is a point-by-point response to the reviewers' comments. We would like to submit this revised manuscript to the “Biomolecules”, and hope it will be published in the journal.
Looking forward to hearing from you soon.
With kindest regards,
Yours Sincerely,
Yizhou Zou.
Response to reviewer 1
Reviewer 1:
The manuscript titled “LILRB4 checkpoint for immunotherapy: Structure, Mechanism and Disease targets”, submitted by Dr Yizhou Zou and his group, is an interesting review of current relevance that attempts to describe the most relevant findings on the role of LILRB4 in diseases; as autoimmune and oncological diseases, as well as its possible use as a therapeutic alternative. Although the specific ligand for LILRB4 is not known and knowledge of the role of LILRB4 is limited, this review describes the advances to date in the knowledge of this important protein. However, there are details that must be addressed and described below to be clearer for the reader.
-There are no references in the first part of the introduction
- Figure abut graphical representation of the LILR family and its domains is suggested.
- In general, there are sections where references are needed, such as in the paragraph on lines 152-161
- The first part of section 2 of LILRB4 and immune tolerance lines 152-180 is too extensive in terms of central and peripheral tolerance. It is suggested to go more directly to the role of LILRB4.
- There are serious errors in the nomenclatures of the cell types of the immune system and the descriptions pf abbreviations are repetitive. (example; Ts, NK, CD8+ Ts)
- The role of LILRB4 in autoimmune diseases such as SLE and IBD is unclear. Describe in greater detail the findings of the role of LILRB4 in these diseases and what is represented in Figure 3 is not very clear according to what is described in the section.
- Figure from section 4.2 is suggested, to help the reader and be clearer in the mechanisms described in the section. The same for the other sections (4.3, 4.4 and 4.5) described.
Point 1:There are no references in the first part of the introduction
Response 1: Thank you very much for your careful scrutiny. Based on your suggestions, we have added the appropriate references to the introduction section.
Point 2:Figure abut graphical representation of the LILR family and its domains is suggested.
Response 2: Your valuable suggestions are greatly appreciated. We have added a diagram of the LILRB family and its structural domains to the paper (see lines 87-92).
Point 3:In general, there are sections where references are needed, such as in the paragraph on lines 152-161
Response 3: Thank you very much for your careful scrutiny. Based on your suggestions, we have scrutinized the content in the article and refined the addition of references.
Point 4:The first part of section 2 of LILRB4 and immune tolerance lines 152-180 is too extensive in terms of central and peripheral tolerance. It is suggested to go more directly to the role of LILRB4.
Response 4: Thank you very much for your comments and professional advice. We have revised the first part of Section II to briefly depict the content on central and peripheral tolerance and to focus the description on the LILRB family and LILRB4 (see lines 175-233).
Point 5:There are serious errors in the nomenclatures of the cell types of the immune system and the descriptions pf abbreviations are repetitive. (example; Ts, NK, CD8+ Ts)
Response 5: Thank you for your careful scrutiny. Based on your suggestions, we have carefully examined and made changes to the naming of immune cells and acronyms in the text to standardize the use of acronyms in the text.
Point 6:The role of LILRB4 in autoimmune diseases such as SLE and IBD is unclear. Describe in greater detail the findings of the role of LILRB4 in these diseases and what is represented in Figure 3 is not very clear according to what is described in the section.
Response 6: Thank you for your valuable input for us. Based on your comments, we have described in detail the role played by LILRB4 in three different autoimmune diseases and have modified Figure 3 in light of the redescription (see lines 308-370).
Point 7:Figure from section 4.2 is suggested, to help the reader and be clearer in the mechanisms described in the section. The same for the other sections (4.3, 4.4 and 4.5) described.
Response 7: Thank you very much for your comments and professional advice. The graphs we started with are for 4.1 content; there are no graphs for 4.2 content. Based on your suggestions, we have revised the diagram in 4.1 and have organized and merged the contents of 4.2 and 4.3 into a single diagram. After supplementation and refinement, the content of a total of 5 figures in the text summarizes for the way LILRB4 works and the role it plays in the disease, so the content of the latter text does not use the graphical representation again.

Reviewer 2 Report
Comments and Suggestions for Authors
The manuscript is generally well-written and informative. However, there are a few suggestions for improvement:
Abstract:
Consider improving the organization for better flow.
Specify potential therapeutic applications more explicitly.
Introduction:
Clarify the term "over-immunization" for better understanding.
Clearly define the abbreviation "LILRB" upon first use.
Improve the transition between immunosuppressive drugs and immune checkpoint proteins.
Structure, Expression of LILRB4, and Its Ligand Molecules:
Break down the detailed paragraph on LILRB4 structure for easier reading.
Briefly explain ITIM (Immunoreceptor Tyrosine-based Inhibitory Motif).
LILRB4 and immune tolerance:
Define abbreviations "Th" and "CTL" the first time they are used.
Clarify whether the inhibition of T lymphocytes and NK cells is in a specific disease context.
Role of LILRB4 in the immune response:
Introduce the abbreviation "APCs" (antigen-presenting cells) upon first use.
Specify cell types when mentioning "Th1 and Th17."
LILRB4 modulates intracellular signaling pathways:
Simplify language for improved readability.
LILRB4 plays its unique role in epigenetic modification:
Clearly define "FTO" upon first use.
Simplify the last sentence for better comprehension.
Regulatory role of LILRB4 molecules in disease:
Clarify whether LILRB4 is a therapeutic target or a marker for autoimmune diseases.
Explain the relevance of highly expressed LILRB4 in "susceptible BXSB/Yaa mice."
Transplantation immune tolerance:
Simplify the language and structure of the last paragraph for better readability.
General Suggestions:
Provide citations for specific findings and studies.
Ensure consistency in terminology and abbreviations.
Conclude with a clear summary of key findings and implications for immunotherapy.
Author Response
Dear Reviewer:
We are very grateful to your valuable comments on this article, which have been helpful in improving the quality of the manuscript. We have carefully considered and responded to each of the suggestions made by the reviewer. The following is a point-by-point response to the reviewers' comments. We would like to submit this revised manuscript to the “Biomolecules”, and hope it will be published in the journal.
Looking forward to hearing from you soon.
With kindest regards,
Yours Sincerely,
Yizhou Zou.
Response to reviewer 2
Reviewer 2:
The manuscript is generally well-written and informative. However, there are a few suggestions for improvement:
Abstract:
Consider improving the organization for better flow.
Specify potential therapeutic applications more explicitly.
Introduction:
Clarify the term "over-immunization" for better understanding.
Clearly define the abbreviation "LILRB" upon first use.
Improve the transition between immunosuppressive drugs and immune checkpoint proteins.
Structure, Expression of LILRB4, and Its Ligand Molecules:
Break down the detailed paragraph on LILRB4 structure for easier reading.
Briefly explain ITIM (Immunoreceptor Tyrosine-based Inhibitory Motif).
LILRB4 and immune tolerance:
Define abbreviations "Th" and "CTL" the first time they are used.
Clarify whether the inhibition of T lymphocytes and NK cells is in a specific disease context.
Role of LILRB4 in the immune response:
Introduce the abbreviation "APCs" (antigen-presenting cells) upon first use.
Specify cell types when mentioning "Th1 and Th17."
LILRB4 modulates intracellular signaling pathways:
Simplify language for improved readability.
LILRB4 plays its unique role in epigenetic modification:
Clearly define "FTO" upon first use.
Simplify the last sentence for better comprehension.
Regulatory role of LILRB4 molecules in disease:
Clarify whether LILRB4 is a therapeutic target or a marker for autoimmune diseases.
Explain the relevance of highly expressed LILRB4 in "susceptible BXSB/Yaa mice."
Transplantation immune tolerance:
Simplify the language and structure of the last paragraph for better readability.
General Suggestions:
Provide citations for specific findings and studies.
Ensure consistency in terminology and abbreviations.
Conclude with a clear summary of key findings and implications for immunotherapy.
Point 1:
Abstract:
Consider improving the organization for better flow.
Specify potential therapeutic applications more explicitly.
Response 1: We sincerely appreciate your serious comments! Thank you for recognizing our work. Based on your suggestions, we have modified the organization of the abstract accordingly and have included a brief description of the potential therapeutic applications of LILRB4 in various diseases (see lines 8-23).
Point 2:
Introduction:
Clarify the term "over-immunization" for better understanding.
Clearly define the abbreviation "LILRB" upon first use.
Improve the transition between immunosuppressive drugs and immune checkpoint proteins.
Response 2: Thank you very much for your comments and professional advice. We defined LILRB when it was first used (see line 56), and we refined the use of the term "over-immunization" and the transition between immunosuppressive drugs and immune checkpoint proteins (see lines 36-45).
Point 3:
Structure, Expression of LILRB4, and Its Ligand Molecules:
Break down the detailed paragraph on LILRB4 structure for easier reading.
Briefly explain ITIM (Immunoreceptor Tyrosine-based Inhibitory Motif).
Response 3: We are deeply grateful for your valuable suggestions. We provide a brief explanation of the ITIM motifs in the text (see lines 106-111), and we have broken down the detailed paragraphs on the structure of LILRB4 to make it easier to read.
Point 4:
LILRB4 and immune tolerance:
Define abbreviations "Th" and "CTL" the first time they are used.
Clarify whether the inhibition of T lymphocytes and NK cells is in a specific disease context.
Response 4: Thank you very much for your careful examination and professional opinion. Based on your suggestions, we have defined acronyms such as Th and CTL where they first appear in the text (see lines 225-226). And, we describe in detail how and in what context the inhibitory effect of LILRB4 on the action of T cells and NK cells occurs (see lines 237-250).
Point 5:
Role of LILRB4 in the immune response:
Introduce the abbreviation "APCs" (antigen-presenting cells) upon first use.
Specify cell types when mentioning "Th1 and Th17."
Response 5: Thank you very much for your careful scrutiny. Based on your suggestion, we introduced its abbreviation, APC, on the first reference to antigen-presenting cells (see lines 149-150). We have revised the content of the article, and for the first occurrence of cells in the text, we have indicated the cell type and introduced the relevant abbreviations.
Point 6:
LILRB4 modulates intracellular signaling pathways:
Simplify language for improved readability.
Response 6: Thank you very much for your professional advice. We have refined the language and optimized the structure of the paragraph in the hope of improving its readability (see lines 269-279).
Point 7:
LILRB4 plays its unique role in epigenetic modification:
Clearly define "FTO" upon first use.
Simplify the last sentence for better comprehension.
Response 7: We are deeply grateful for your valuable suggestions. Based on your suggestions, we have clearly defined FTO and optimized the language and structure of the last sentence of the paragraph (see lines 284-292).
Point 8:
Regulatory role of LILRB4 molecules in disease:
Clarify whether LILRB4 is a therapeutic target or a marker for autoimmune diseases.
Explain the relevance of highly expressed LILRB4 in "susceptible BXSB/Yaa mice."
Response 8: Thank you very much for your professional opinion. Based on your suggestions, we have made changes to the relevant parts of the article. We describe in detail the role of LILRB4 in autoimmune diseases and explain the relevance of LILRB4 to BXSB\Yaa mice in autoimmune disease studies (see lines 308-333).
Point 9:
Transplantation immune tolerance:
Simplify the language and structure of the last paragraph for better readability.
Response 9: We are deeply grateful for your valuable suggestions. Based on your suggestions, we have optimized the language and structure of the last paragraph (see lines 419-432).
Point 10:
General Suggestions:
Provide citations for specific findings and studies.
Ensure consistency in terminology and abbreviations.
Conclude with a clear summary of key findings and implications for immunotherapy.
Response 10: Thank you very much for your professional opinion. We have added good specific references wherever there are specific findings and research in the text. We carefully checked the full text to ensure consistency of terminology and abbreviations. In the final concluding section of the article, we explicitly summarize the main findings to date for the LILRB4 molecule and the implications of the LILRB4 molecule for immunotherapy (see lines 602-633).

Round 2
Reviewer 1 Report
Comments and Suggestions for Authors
The manuscript titled “LILRB4 checkpoint for immunotherapy: Structure, Mechanism and Disease targets”, submitted by Dr Yizhou Zou and his group, worked on the suggested changes improving the quality of the manuscript